# Influence of the Type of Plastic and Printing Technologies on the Compressive Behavior of 3D-Printed Heel Prototypes

**DOI:** 10.3390/ma16051930

**Published:** 2023-02-26

**Authors:** Edita Gelaziene, Daiva Milasiene

**Affiliations:** Department of Production Engineering, Faculty of Mechanical Engineering and Design, Kaunas University of Technology, 51424 Kaunas, Lithuania

**Keywords:** 3D printing, additive manufacturing, polymeric heels, compressive loads, FEM simulation

## Abstract

In this study, the possibility of using modern AM technologies to produce designed heels for personalized orthopedic footwear with a medium heel was explored. Seven variants of heels were produced using three 3D printing methods and polymeric materials with different natures: PA12 heels made using the SLS method, photopolymer heels made using the SLA method, and PLA, TPC, ABS, PETG, and PA (NYLON) heels made using the FDM method. A theoretical simulation with forces of 1000 N, 2000 N, and 3000 N was performed in order to evaluate possible human weight loads and possible pressure during orthopedic shoe production. The compression test of the 3D-printed prototypes of the designed heels showed that it is possible to replace the traditional wooden heels of hand-made personalized orthopedic footwear with good-quality PA12 and photopolymer heels made using the SLS and SLA methods, but also with PLA, ABS, and PA (NYLON) heels printed using a cheaper FDM 3D printing method. All of the heels made using these variants withstood loads of more than 15,000 N without damage. It was determined that TPC is not suitable for a product of this design and purpose. Due to its greater brittleness, the possibility of using PETG for orthopedic shoe heels must be verified by additional experiments.

## 1. Introduction

Today, manufacturing technology plays a vital role in economic growth, industrial applications, and scientific and technological progress. As technology advances rapidly, manufacturing technology has evolved from the era of traditional manufacturing to an era of intelligent, highly efficient, and sustainable manufacturing. Now widely known as ‘rapid prototyping’ (RP) or ‘additive manufacturing’ (AM), 3D printing was invented in the late 1980s. Three-dimensional printing has quickly gained considerable attention and has developed as a new manufacturing technology [1]. Recently, the usage of 3D printing has been expanding very rapidly due to the joint interest from both industry and research communities. The successful improvement of AM technologies allows ever-improving quality prints to be produced quickly and relatively inexpensively. This method of manufacturing is interesting in various areas due to its successful results in the production of complex-shaped parts, saving materials and time thanks to its rapid production process [2,3,4]. Additive manufacturing technologies allow the manufacture of very complex shapes or geometries and greatly expand the possible design solutions [5]. Such advantages as lower amounts of material waste, lower post processing, and obviously lower production costs of even complex-shaped parts make 3D printing technologies the sustainable technologies of the future. In addition to important sustainable aspects, there is the possibility to recycle and reuse plastic, reducing pollution [6].

Wide-ranging design possibilities, lower cost, and much faster achievement of the final result have led to 3D printing technologies gaining great importance in the fashion world in recent years. They are used in various fields of application, and the production of footwear is no exception. Additive manufacturing technologies open up a novel way to produce new-fashioned footwear by integrating new plastic materials and digital production [7]. Three-dimensional printing has a layer-by-layer production strategy that fundamentally differs from conventional subtractive manufacturing technology [8]. The application of 3D printing can play a significant role in the field of future footwear [9].

Advances in 3D printing technologies have facilitated the personalization of orthopedic products, rapid prototyping, correction, and production. FDM 3D printing technology and PLA filaments have been successfully used to manufacture foot orthoses (FOs) for people with flexible flat feet (flatfoot). After studying the mechanical properties and biomechanical effects of 3D-printed FOs in individuals with problematic feet, scientists concluded that, in this low-cost and rapid way, manufactured custom FOs provided sufficient support to correct foot abnormalities [10]. Researchers from Croatia created and tested 3D-printed heels from ABS material manufactured using FDM technology in two different orientations. It was found that flexural strength had higher values when heel prototypes were printed in the vertical position. The authors argue that additive manufacturing enables producing unique personalized heels but at the same time obtaining certain mechanical properties with selected parameters and orientations of 3D printing [11]. Lim et al. [12] developed a design methodology for lightweight (3D) tetrahedral meshes. The designed honeycomb structure and the acrylic photopolymer (3DK83I) cubes printed with a DLP printer (IM96) were subjected to a compression test, where it was found that these cubes could withstand a load 10,000 times greater than their own weight. These lightweight mesh structures were adapted to make replacement heels for women’s shoes. After performing a finite element analysis of the heels with the Ansys program, they found that the maximum stress occurred at the rear bottom of the heel, which did not exceed the allowable stress limit for the tetrahedral model mesh.

Additive manufacturing technologies offer new opportunities and creative solutions for creating customized production. This is particularly important to improve orthopedic products to meet patients’ needs. One of the most popular areas of orthopedics is footwear, which is made for people with foot problems such as foot pain and deformity, and movement disorders. In custom footwear manufacturing, the most important purpose of the production process is to produce shoes that perfectly fit the dimensions of the foot and match the desired model [13]. Orthopedic shoes are prescribed to patients who suffer from various illnesses, including diabetes, rheumatic diseases, and degenerative foot diseases. Orthopedic shoes are made by manually applying standardized shoe elements that are adjusted each time depending on the foot problem. It is difficult to achieve design solutions and an excellent aesthetic appearance because changes in height, shape, and angle of inclination are made manually, and changes are visible, often resulting in an unattractive design. Van Netten et al. [14] studied the reasons that determine the patient’s decision to use custom-made orthopedic footwear. As the most important factor, the patients indicated an improvement in walking comfort, while the lack of design of the orthopedic shoes they accepted as a compromise. With the development of new 3D printing technologies, there is an opportunity to incorporate modern design solutions into areas such as orthopedic footwear that lack design and aesthetic sensibility. These technologies make it easy and fast to create and adapt individual footwear parts, such as soles or heels, to a particular patient, immediately assessing foot problems and making the necessary adjustments. On the other hand, despite these advantages, the application of 3D printing in the footwear industry is still limited because this technology currently does not enable a mass application that includes high-quality material behavior and geometric shape design to meet the high demands of customers in the market [7].

Modern 3D printing is an advanced technology that, with the help of a computer-aided design system, can help solve some of these problems. Three-dimensional (3D) printing allows one to create physical forms of almost any complexity (stored in a CAD model) [15]. There are about 20 known RP technologies in existence. The most commonly used techniques are fused deposition modeling (FDM) by means of a plastic filament, powder-based selective laser sintering (SLS), and stereolithography (SLA), during which a liquid resin is used. Due to the lower cost, higher printing speed, high strength and robustness, non-toxicity, variety of materials, and flexibility of application, FDM is the most preferred AM technology for polymer and composite materials [16]. This most widely used simple-operation 3D printing method was patented by Crump in 1988. FDM is based on the process of melted thermoplastic material extrusion through the nozzle [17]. The easy change of materials, low working temperature, relatively cheap maintenance, and compact size give an edge to this technology. On the other hand, commercial 3D printers suitable for larger-scale production can often only print using ABS or PLA; therefore, there are opinions that this 3D printing method has a limited range of appropriate materials [18]. The parameters of the FDM process have a great influence on the properties of the final product, such as porosity, strength, and surface finish [19].

For FDM printing, thermoplastic filaments from polylactic acid (PLA), polyamide (NYLON), acrylonitrile butadiene styrene (ABS), polycarbonate (PC), polyether-ether-ketone (PEEK), and others are used. The most popular among them are PLA and ABS [20]. ABS filaments exhibit high melt strength and durability, adequate mechanical properties, and toughness; simultaneously, they are quite easy to print and therefore are one of the most common filaments for FDM printing [21].

The intensive development of new technologies in recent years has caused great interest in research on the influence of the technological parameters of 3D printing on the mechanical behavior of manufactured parts. Many works have been published on the study of the influence of various factors (the nature of thermoplastic materials, the thickness of the layer deposited by the nozzle, the layer’s orientations, the raster angle, the air gap between two adjacent deposited filaments, etc.) on the mechanical properties of printed products. Wu et al. [18] studied the influence of deposited layer thickness and different orientation raster angles on the mechanical properties of samples printed from two types of plastics: ABS and polyether-ether-ketone (PEEK) [18]. Banjanin et al. [22] investigated the mechanical tensile and compression properties of FDM-printed parts from PLA and ABS thermoplastic material. According to the results of the compression test, the PLA samples showed better consistency with regard to mechanical properties compared to the ABS specimens.

The TPU material is also a fairly commonly used 3D printing material that possesses flexible properties, elasticity, shock resistance, and thermoplasticity. This material has been used to explore the possibility of producing 3D-printed supportless closed-cell lattice structures, which behave as nonlinear springs with great energy absorption properties. Kumar et al. [23] were inspired by nature: the idea for unit cell design came from the sea urchin’s morphology. Such structures can be applied to productions with special functional requirements—for example, in the manufacturing of lightweight midsole shoes.

Sousa et al. [24] studied the possibility of replacing conventional dental guards for athletes with new 3D-printed ones. They investigated the impact-energy-absorbing capability of printed samples from such materials as PLA (recycled plastic for food packaging), thermoplastic polyurethane (TPU), high-impact polystyrene (HIPS), and poly(methyl methacrylate) (PMMA). In this work, recycled PLA, PMMA, and HIPS had a higher transverse impact strength than TPU and ethylene–vinyl acetate (EVA) (standard material). Lee et al. [25] studied the possibility of producing personal protective equipment by means of 3D printing from thermoplastic polyurethane (TPU). TPU was chosen as an easily available material on the market, a flexible material that can protect against shock and improve comfort during movement. It was obtained that all the created and printed samples absorbed a total energy of at least 5 J and confirmed that TPU material could be used to create 3D-printed personal protective equipment (such as knee protectors).

Material extrusion 3D printing was also used for processing shoe lasts from thermoplastic polymers. In order to reduce the weight and production time, Amza et al. [13] designed shoe lasts with zones of different resistance to external forces. In this work, acrylonitrile butadiene styrene (ABS), polylactic acid (PLA), and polyethylene terephthalate glycol (PETG) were used for the production of shoe last prototypes for testing in a real shoe-production setting.

Another widely used method is selective laser sintering (SLS). SLS is a technology in which polymer powders are processed into solid 3D models by exposing them to a high-power ultraviolet laser. While FDM dominates the broad consumer market due to the relatively low cost of 3D printers using this method, SLS technology is more commonly used in professional and industrial production due to its high quality and technical capabilities, which offset the higher cost of the devices [26]. The process parameters of the SLS process have a great influence on the mechanical behavior of the printed models, as this technology is based on heat input from the sintering laser, and the density of prints depends on the laser power and the process parameters, such as the heating temperature of the powder layers [27].

Among the polymer materials used in SLS, polyamide-12 (PA12) leads the market. This thermoplastic semicrystalline material is characterized by stable processing and slight crystallization shrinkage. In order to achieve the best quality of PA12 SLS-printed products, many studies have been conducted examining the influence of various factors of the sintering process on the mechanical properties of prints during tests for stretching, creep, compression, or fatigue [28,29,30,31,32]. El Magri et al. [33] investigated the influence of SLS parameters such as the hatch orientation and laser power on the properties and mechanical behavior of 3D-printed PA12 standard samples. They noticed that the optimal level of results was obtained when the higher laser power (95%) and parallel/perpendicular hatching were used. When the laser power reduced to 75%, the mechanical properties and microstructure quality were reduced [33]. Some research has been performed to explore the possibility of using unrecycled polyamide-12 (PA12) powder waste, thereby reducing both environmental pollution and production costs [34,35,36,37,38]. The properties of SLS-printed PA12 samples were also tested by simulating the possible conditions of the application of the products. Hooreweder et al. [39] found that the density of the SLS-printed PA12 components is a main factor in their fatigue life. In this work, a mixture of virgin and previously used unsintered PA12 powder (50%/50%) was used. The researchers found that the lower density of the printed samples indicated a higher amount of unfused powder particles, increasing the chance of crack initiation.

One more well-known AM technology is stereolithography (SLA), the earliest and first method of 3D printing that uses a UV laser as the light of a photopolymer to cure a polymer resin. SLA is a nozzle-free method; therefore, there are no material clogging problems. After printing, the unreacted resin can be reused [40]. The materials used for this AM technology are liquid resins of photosensitive thermoset polymers, whose chemical properties are very important. Usually, they are acrylic and epoxy resins. During photopolymerization, the liquid resin is converted into solid matter by means of a chemical reaction in the presence of ultraviolet (UV) light. For the SLA process, the rheological properties and melting temperature of the liquid resin are very important, as well as the curing kinetics that determine the speed of production but at the same time must ensure enough time for good interlayer adhesion [41]. Miedzińska et al. [42] investigated the compression strength of SLA-printed cubic samples from two types of resin under high and low strain rates. The obtained results showed that both ‘tough’ and ‘clear’ photocurable resins demonstrated elastic–plastic behavior, and the printed specimens did not show any cracks after 50% of the obtained strain. Pasquale et al. [43] investigated the influence of the SLA process parameters (polymer transparency, port-curing time, and hatching direction) on the topological and mechanical properties of biomedical polymers. The authors found that the polymer strongly influenced the geometrical tolerances of the prints; the mechanical properties were affected moderately.

Due to the increasing spread of modern AM technologies in various fields, a number of comparative review articles have been published, analyzing the technological processes of different 3D printing methods and used materials, examining the pros and cons of different methods, and problems relating to sustainability and recycling [41,44,45,46]. Yaragatti and Patnaik [45], in their review, briefly explained the techniques of different additive manufacturing (AM) methods, using polymeric materials, and the opportunities to apply AM for the manufacture of polymer composites, the sustainable AM of polymers, green composites, and polymer recycling. Agrawal [46] reviewed the materials used for the three most common additive manufacturing methods: FDM, SLS, and SLA. The sustainable material selection was performed according to the criteria chosen by the author. The review was intended to help in the new product development stage, with a better choice of materials for 3D-printed products to support cleaner production.

Quite a lot of researchers have prepared 3D-printed specimens in several different ways and compared the anisotropy, mechanical properties, and cost–quality ratio of the prints. Szykiedans and Credo [47] compared the mechanical properties (tensile strength and elastic modulus) of FDM and SLA 3D-printed samples with the application of low-cost printers. The results showed a significant anisotropy of FDM 3D-printed components. It is determined that when a small amount of molten material is extruded from the nozzle to form layers, it hardens immediately. The outer fibers of the printed part are much better fused than the inner layers.

Due to the advantages, and despite the disadvantages, of additive manufacturing (AM) 3D printing methods, the use of this rapid production technology is increasing in various fields of life. Plenty of research works involve efforts to improve the technologies of 3D printing and to obtain the best result for printed products, to improve sustainability and pollution waste reduction [45].

Kafle et al. [41] reviewed 272 articles and separated the main advantages and limitations of the three most commonly used AM methods (FDM, SLS, and SLA). They characterized FDM as a fast, low-cost, 3D printing method that can fairly accurately produce medium-complexity design parts using a wide variety of materials. Its deficiencies are poor surface finish, medium resolution, and the requirement of support structures. SLS was marked as the method with the best accuracy, the greatest design freedom, and a better surface finish than in the case of FDM. A major plus is that here no support structures are required; however, SLS printing lasts longer and is more expensive than FDM. In the case of SLA, the authors emphasized high accuracy, high resolution, and a wide range of functional applications. It is quite easy to use, and the best surface-finish quality can be obtained. As a negative of SLA, the limited choice of suitable materials and the high maintenance cost were mentioned.

This paper presents an ongoing study of opportunities to expand the application of modern AM technologies to the production of customized orthopedic footwear. After reviewing the experience of other researchers, it was decided to test the possibilities of using the three most commonly used AM methods (FDM, SLS, and SLA) and different materials for the manufacture of designed heels for orthopedic shoes.

## 2. Materials and Methods

### 2.1. Object of Study

Orthopedic footwear is produced in a variety of styles and models, but one of the most popular models is shoes with heels. In the usual method, standard-type wooden heels of the right size (Figure 1a) are adapted when the deformations of the foot are slight. In the case of strong foot deformations, the heels are made of a wood blank by hand by adapting the heel model and the size according to the customer’s foot measurements.

Due to the intensive development of modern AM technologies in orthopedics, the search for opportunities to apply faster and cheaper 3D printing technologies is also being conducted. Several different designs and heights of heels for orthopedic shoes have been designed using SolidWorks software. In this work, low-height heels (3 cm) were studied (Figure 1b).

### 2.2. Materials and 3D Printing Methods

In order to successfully integrate AM technologies into the production of orthopedic footwear, it is necessary not only to create a new design but also to modernize the entire production process with the latest digital technologies (Figure 2). Precise 3D scanning of deformed or otherwise damaged feet plays an important role in ensuring a high accuracy of personalized 3D-printed parts. The ability to link these two modern digital technologies not only enables the design of the bottom parts of footwear, such as soles, heels, orthopedic insoles, or a foot orthosis (arch support), exactly according to the specific deformation of the foot but also gives more freedom in the choice of design shapes and colors.

The prototypes of the designed orthopedic shoe heels investigated in this work were produced by means of three AM methods: FDM, SLS, and SLA, using seven polymeric materials of different natures: thermoplastic copolyester (TPC), polylactic acid (PLA), acrylonitrile butadiene styrene (ABS), polyamide-12 (PA12) powder, polyethylene terephthalate glycol (PETG), photopolymer, and polyamide filaments (NYLON). The characteristics of the materials used and the corresponding 3D printing methods are presented in Table 1. Three-dimensional FDM printing was performed using the printer ‘Layercube Custom Delta’, SLS ‘EOS 396′, and SLA—‘Form 2′. In this work, a mixture of virgin and used unsintered PA12 powder (50%/50%) was used. Repeated PA12 powder was used no more than seven times.

### 2.3. Static Simulation

Static analysis of the created heels was performed using the ‘Simulation’ function of the SolidWorks program with the finite element method (FEM). The simulation was carried out by rigidly fixing the model on both the straight and convex sides. The models were loaded with 1000 N, 2000 N, and 3000 N forces selected according to the possible loads of human weight (up to 100 kg) and according to the possible pressure during orthopedic shoe production. The parameters of the finite element mesh are shown in Figure 3.

### 2.4. Compression Test

The compression test of the prototypes of the designed heels was performed using the universal testing machine Tinius Olsen H25KT (Salford, United Kingdom) (Figure 4a) with a special compression supplement (Figure 4b). The compression speed was 5 mm/min. The 50 N preload was set for all tests. A special punch according to the concave top of the heel was formed from steel-reinforced epoxy putty ‘Abro Steel’.

## 3. Results and Discussion

### 3.1. Virtual Static Simulation

A theoretical compression simulation was performed to model the response of orthopedic shoe heels designed from different polymeric materials to possible loads during wear or shoe manufacturing processes. A total of 21 simulation variants (7 different materials and 3 load variants) were performed. It was observed that the distributions of von Mises stresses and displacements in the spatial construction of the heel do not depend on the properties of the selected polymeric material. The simulation images for the seven material variants are visually the same, and so in Figure 5 and Figure 6, the examples of the obtained results are presented.

During real shoe production, the heel can be pressed by higher loads compared to the load from the possible weight of the human body. For this reason, simulations with higher forces of 2000 N and 3000 N were performed. As can be seen from Figure 5, by increasing the pressure force from 1000 N to 3000 N, the mapping of the distribution of stress and displacements does not change. Analogous results were obtained with other investigated materials.

It should also be mentioned that during the theoretical simulation, when the materials and their properties changed, the values of stresses and displacements also changed (Figure 7), but the location of the zones of greatest impact was the same in all cases. As an example for comparison, Figure 6 presents the results of the static simulation of heels when a material of a different nature, photopolymer, was chosen. Repetitive areas of the greatest impact in all samples are defined by a red oval.

It was determined that the highest stresses occur on the back side of the heel in the narrowest places of concave pillars, and the values of maximum displacements are predicted in the upper edge area for both the stiffest materials (PA12, photopolymer) and softest material (TPC). However, the consequent stresses are significantly lower than the ultimate strength of all tested materials (Table 1) or yield strength values used in the theoretical simulations works of other researchers: 20 MPa for ABS [48], 45 MPa for PLA [49], 49.7 MPa for PETG [50], and 82 MPa for NYLON [51]. Therefore, no plastic deformation will occur in the areas with the greatest impact of heels.

Analysis of the influence of material nature on the static simulation results showed that the highest stresses in the dangerous heel zone are predicted for NYLON (3.816 MPa at 1000 N load force), but the results obtained for PA12 and PLA are also very close (3.741 MPa and 3.739 MPa, respectively) (Figure 7a). Slightly lower stresses are also predicted for ABS (3.674 MPa) and PETG (3.631 MPa). The lowest stresses were obtained for the TPC polymeric material (3.040 MPa).

In all cases of the seven examined materials, the increase in compressive force from 1000 N to 3000 N triples the maximum stresses in the dangerous zones of the created heels, while the tendency of the differences between different materials remains the same.

The obtained results of theoretic simulation show that the spatial design of the designed heels for orthopedic shoes is the main factor that determines their resistance to compressive loads up to 3000 N, while the influence of the properties of the used materials is significantly smaller.

A different situation was obtained when the results of the maximum displacements were analyzed. In this case, the properties of the materials greatly influenced the values of maximum displacements. As seen in Figure 7b, the largest displacement values are predicted during the compressive load simulation for the TPC heel. In all three cases of load force, displacement results were obtained that were more than nine times larger than in the case of the NYLON heel (0.625 mm, 1.255 mm, 1.893 mm and 0.068 mm, 0.136 mm, 0.205 mm, respectively). This can be attributed to the specific composition of this material that combines the properties of thermoplastics and rubber materials. In all other cases, the displacement values are lower and do not exceed 0.15 mm.

In the case of photopolymer and PA12, the displacement values are predicted to be practically equal, whereas the smallest values of displacement are obtained in the case of the PLA heel.

### 3.2. Compression Test

Seven variants of orthopedic footwear heel prototypes from materials of different natures and printed using the three most commonly used methods were tested in a compression test. The obtained results are presented in Figure 8, Figure 9, Figure 10 and Figure 11. The preload of the compression test was 50 N, and it was repeated five times. Figure 8 shows the inherent curves for each group of samples.

As expected, the experimental results differed much more between the different heel variants than in the theoretical simulations. As is known from the experience of other researchers, the AM production method has a major influence on the mechanical properties of the 3D-printed parts. Therefore, the results of the heels printed using the SLS method from PA12 and printed from photopolymer using the SLA method clearly stood out.

As can be seen from the presented curves (Figure 8), these variants of heel prototypes did not reach the first bending or the other break in the spatial structure of the heel’s moment until the maximum limit of the force sensor of the testing machine.

For this reason, no damage or changes in visual view were observed in the specimens after the test when the load was removed (Figure 9). During the test, the compressing procedure was stopped when the limit of 24,800 N was exceeded, so it can be said that the possible loads of such heels during the production of orthopedic shoes and their wearing would not be dangerous.

Some deformities in the feet change quickly enough that orthopedic shoes must be changed quite often. Since production by means of the SLS and SLA methods is more expensive, five different types of polymeric materials were tested in search of possibilities to produce customized orthopedic footwear heels in a cheaper way. In this case, the heels from TPC, PLA, ABS, PETG, and PA (NYLON) were printed using the same FDM method, so the compression resistance was determined according to the properties of the used polymeric material. The results of the compression experiment showed that the FDM-printed heels from PLA, ABS, and PA (NYLON) were also strong enough and resistant to high compression loads.

The moment of the first break in the spatial structure of the heels in the compression curve of PLA heels was determined after reaching a force of more than 20,000 N, more than 19,000 N in ABS heels, and more than 16,000 N in PA (NYLON). The distribution of these results corresponded to the distribution of the modulus of elasticity of PLA, ABS, and PA (NYLON) (Table 1). It is also consistent with other researchers’ claims that ABS material exhibits greater elasticity than PLA [22]. As predicted, once the critical force is reached, bending and damage occur in the narrowest places of concave pillars in the structure of the heel. With further compression, the areas of damage become apparent (Figure 10).

Of these three variants, the PLA heel prototypes were characterized by the highest resistance to compressive loads. As can be seen from the presented experimental curves, up to the limit of the compression force of 17,000 N, the load–displacement curve of the PLA heel printed using the FDM method practically coincides with the curves of the PA12 and photopolymer heels printed using the SLS and SLA methods. It can be presumed that the further behavior of the PLA heels as a spatial system during compression is determined by the specificity of the FDM-method printing process when the formed volume is filled not with a solid sintered mass but with fused plastic filament layers. The fused filament system has less homogeneity, and so it begins to break down under compression at lower loads. In the case of PLA, ABS, and NYLON FDM-printed heels, cracks were caused by the separation of fused filaments (Figure 11). An analogous nature of the damage of FDM samples is mentioned in the work of other researchers [52].

On the other hand, an experimental study showed that the TPC polymeric material as a 3D-printed heel is not suitable for replacing wooden heels. This material is too soft, and so the concave pillar elements of the heel design begin to bend under a compression load of about 2000 N without reaching the required limit forces of 3000 N. The TPC material is the most elastic of all tested materials. As can be seen in Figure 12a, after the compressive load is removed, the heel recovers largely like rubber parts. This material was tested because it is suitable for the production of other footwear parts; on the other hand, the theoretical maximum displacement was less than 1 mm under a compressive load of 1000 N. The experimental compression test allowed us to determine that this polymeric material is too soft and too weak to replace wooden heels in orthopedic shoes with medium–high heels.

Although PETG heel prototypes withstand the limit of 3000 N under compression and fail only after reaching a compression load of 7000 N, this material was also more brittle than the others because it cracked not only in the narrowest locations of the concave pillars but also across the fused plastic filaments in the vertical direction (Figure 12b). By analyzing the PETG heels’ compression curves, it was found that after reaching a load of 1000 N, which simulates the possible weight of a person, the deflection values of the PETG samples were obtained close to the results in the case of ABS and NYLON heels (about 1.00 mm). To make sure that the PETG material could be used for 3D-printed heels for orthopedic shoes instead of wooden heels, more research needs to be done—for example, determining the resistance to cyclic loads.

In summary, the results of this study clearly show that it is possible to replace traditional wooden heels in orthopedic footwear with PLA, ABS, and PA (NYLON) heels printed using a cheaper FDM 3D printing method.

## 4. Conclusions

This paper is part of a study on the possibilities for the modern optimization of the provision of custom-made footwear for patients with foot deformities. The application of modern AM technologies in the field of orthopedics is very important to speed up and make the production of individual orthopedic shoes cheaper when the deformities of the feet not only change rather quickly due to the disease but also develop differently in the feet of the same person. The important thing is that the integrated application of modern digital technologies such as the 3D scanning, 3D design, and 3D printing complex allows one to achieve the best results in the modern development of personalized manufacturing and, at the same time, provide the patient with greater options for modern design and choice according to their individual tastes.

In this work, prototypes of the original designed heel created for orthopedic shoes were produced by means of the three most commonly used 3D printing methods: FDM, SLS, and SLA. Seven different polymeric materials were chosen to produce the designed heel prototypes: thermoplastic copolyester (TPC), polylactic acid (PLA), acrylonitrile butadiene styrene (ABS), polyamide-12 (PA12) powder, polyethylene terephthalate glycol (PETG), photopolymer, and polyamide filaments (NYLON).

The resistance to compressive loads is very important for such products, and so a theoretical static load simulation was performed first, and then the printed prototypes of the designed heels were experimentally tested in a compression test. The theoretical study showed that, when the compression force varies up to 3000 N, the location of the zones of maximum impact in the spatial model of the heel does not depend on the type of material chosen and is instead determined by the geometric dimensions of the design elements.

The experimental study showed that the 3D printing method has a significant influence on the resistance of the manufactured prototypes to compressive loads, so heel prototypes made of PA12 (SLS) and photopolymer (SLA) withstand compressive loads of more than 20,000 N without visible change. It should also be noted that it has been found that TPC materials are not suitable for such products. Further experiments are needed to verify that PETG materials can be used to replace wooden heels in orthopedic shoes.

In summary, five variants of the investigated created heel prototypes are perfectly suitable for further product development: PA12 heels (SLS method), photopolymer heels (SLA method), and PLA, ABS, and NYLON heels (FDM method).

Continuing this work, further research is needed to investigate the nature of the influence of the polymeric material and 3D printing technique on the resistance of the printed heel prototypes to cyclic compression loads. Furthermore, further research will delve into the possibility of changing the geometric dimensions of the design elements and also will investigate the influence of the nature of polymeric materials on the compressive resistance and resistance to the cyclic compressive loads of concave pillars.

## Figures and Tables

**Figure 1 materials-16-01930-f001:**
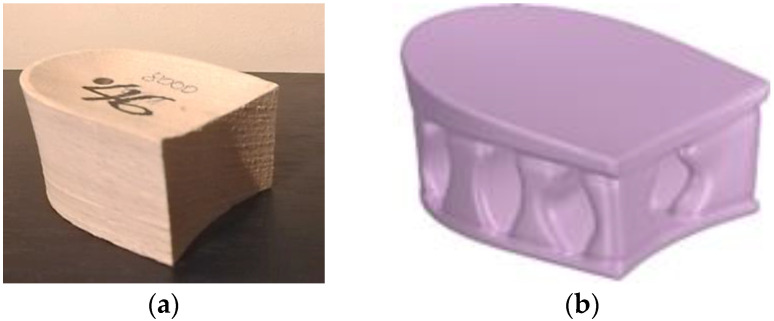
Orthopedic heel: (**a**) standard wooden heel; (**b**) designed heel model.

**Figure 2 materials-16-01930-f002:**
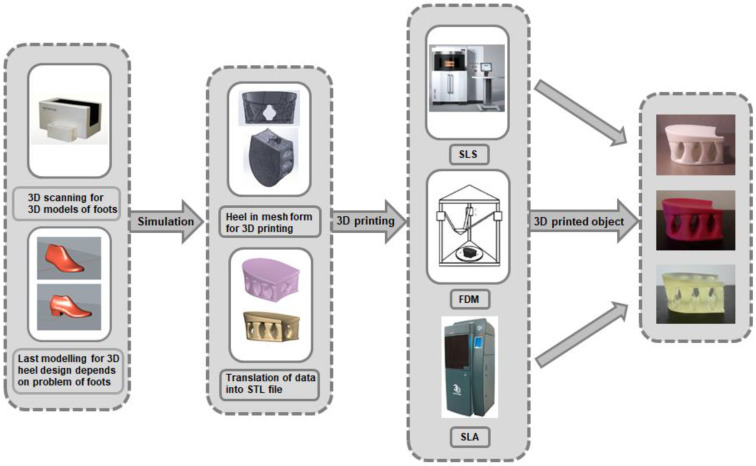
Modern digital technologies in the manufacturing of orthopedic footwear.

**Figure 3 materials-16-01930-f003:**
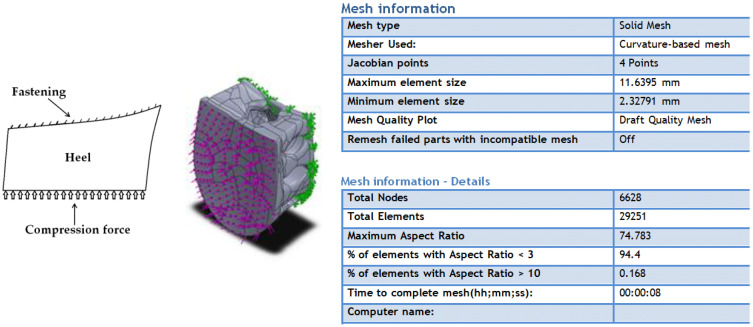
Parameters of the static simulation of designed heels.

**Figure 4 materials-16-01930-f004:**
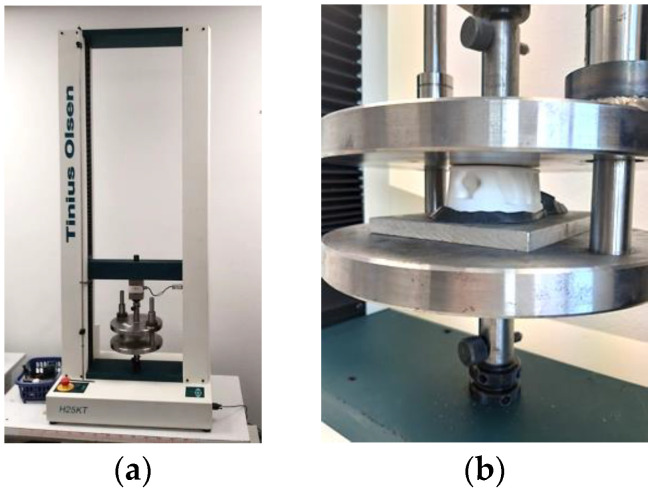
Compression test equipment: (**a**) universal testing machine Tinius Olsen H25KT; (**b**) special compression supplement.

**Figure 5 materials-16-01930-f005:**
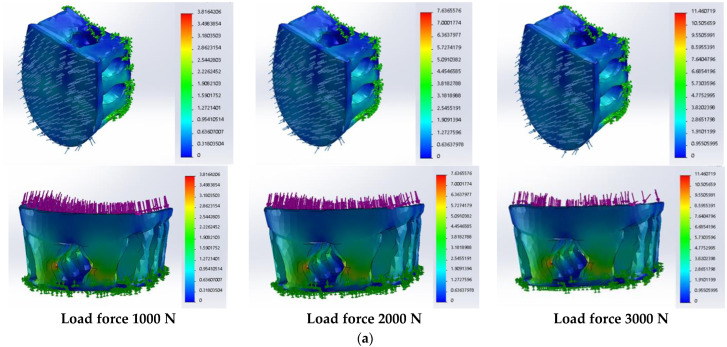
Results of the static analysis of designed orthopedic shoe heels from NYLON: (**a**) distribution of the von Mises stress, MPa; (**b**) distribution of the displacements, mm.

**Figure 6 materials-16-01930-f006:**
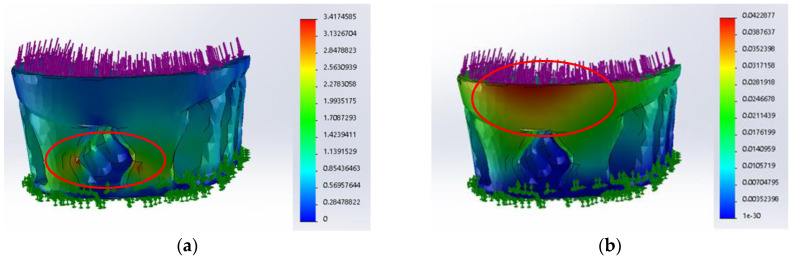
Results of the static analysis of designed orthopedic shoe heels from photopolymer: (**a**) distribution of the von Mises stress, MPa; (**b**) distribution of the displacements, mm. The most dangerous area is circled in red (load force 1000 N).

**Figure 7 materials-16-01930-f007:**
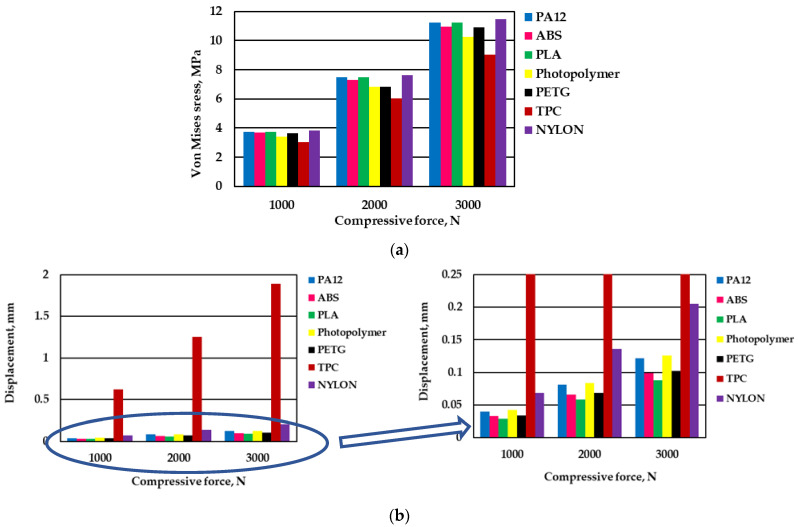
The comparison of results of the static analysis of heels from various materials: (**a**) von Mises stress, MPa; (**b**) displacement, mm.

**Figure 8 materials-16-01930-f008:**
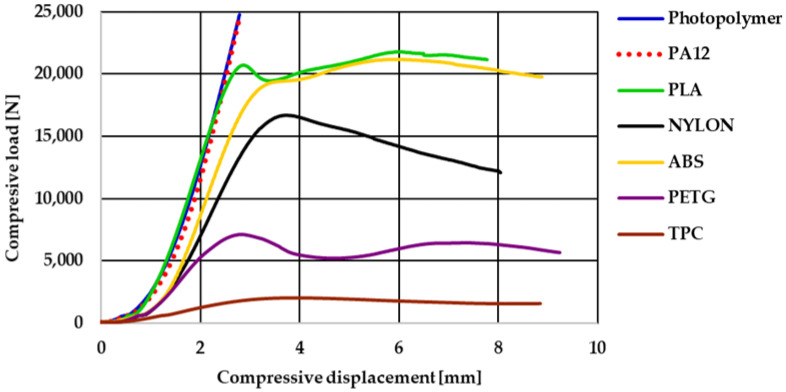
Heel compression test load–displacement curves.

**Figure 9 materials-16-01930-f009:**
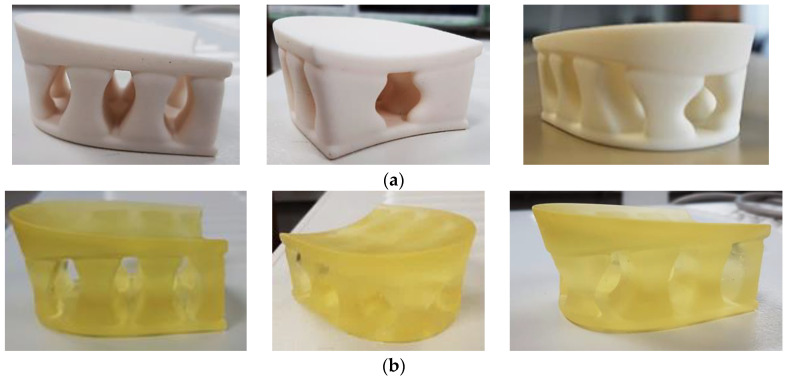
The samples of heel prototypes after compression test: (**a**) heels printed by the SLS method from PA12; (**b**) heels printed from photopolymer by the SLA method.

**Figure 10 materials-16-01930-f010:**
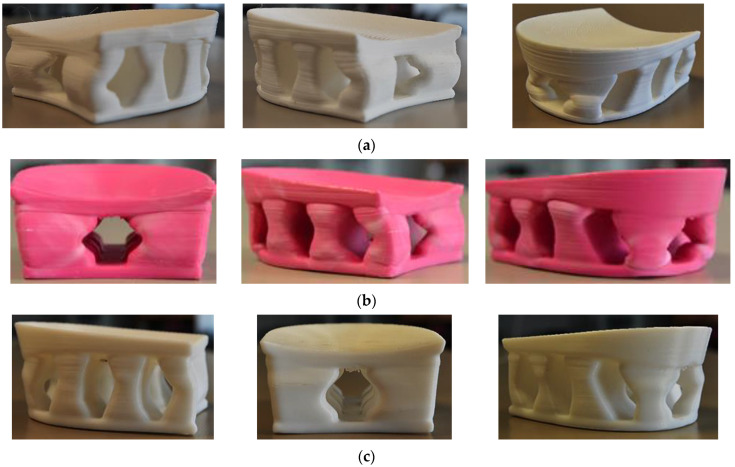
The samples of FDM-printed heel prototypes after compression test: (**a**) heels from PLA; (**b**) heels from ABS; (**c**) heels from PA (NYLON).

**Figure 11 materials-16-01930-f011:**
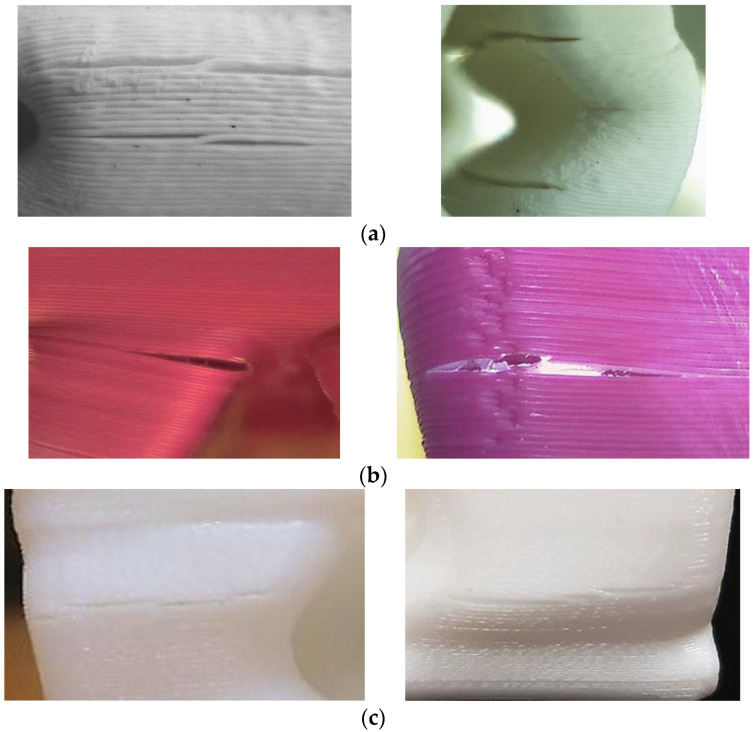
Images of damage between the fused plastic filament of FDM-printed heels: (**a**) heels from PLA; (**b**) heels from ABS; (**c**) heels from PA (NYLON).

**Figure 12 materials-16-01930-f012:**
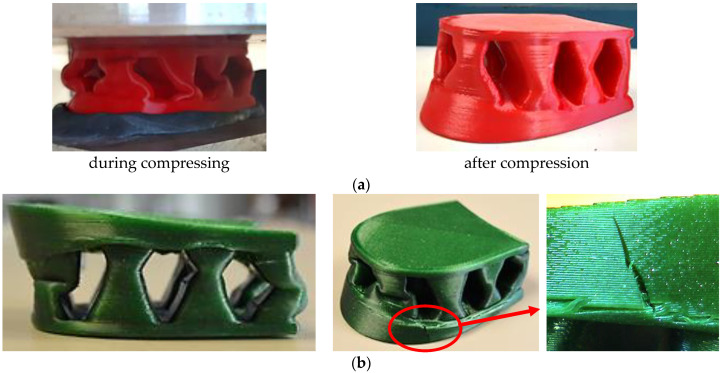
The samples of FDM-printed heel prototypes: (**a**) heels from TPC during and after compression test; (**b**) heels from PETG after compression test.

**Table 1 materials-16-01930-t001:** Characteristics of applied materials and used additive manufacturing methods.

Material	AM Method	Characteristics *			
		Module of Elasticity, Mpa	Poisson Coefficient	Density, g/cm^3^	Ultimate Strength, MPa
PA12 powder	SLS	1650	0.35	0.93	48
PETG	FDM	1940	0.4	1.27	50
PA filament NYLON	FDM	1000	0.3	1.157	60
TPC	FDM	95	0.48	1.14	24
ABS	FDM	1900	0.39	1.02	38
PLA	FDM	2300	0.35	1.30	35.9
Photopolymer	SLA	1500	0.45	1.12	50

* All values are provided in the technical documentation of producers.

## Data Availability

The data presented in this study are available on request from the corresponding author.

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
