# Peer review of "Influence of the Type of Plastic and Printing Technologies on the Compressive Behavior of 3D-Printed Heel Prototypes"

_materials, 2023, doi:10.3390/ma16051930_

Round 1

Reviewer 1 Report

This manuscript fabricated orthopedic shoes using three additive manufacturing technologies (FDM, SLS, SLA) and seven different materials: thermoplastic copolyester (TPC), polylactic acid (PLA), acrylonitrile butadiene styrene (ABS), polyamide 12 (PA12) powder, polyethylene terephthalate glycol (PETG), photopolymer, polyamide filaments (NYLON). The authors then performed simulation to study the stress and displacement distribution of these seven materials during compressing test. In addition, the authors also performed experimental compression test of these seven materials. The manuscript needs to address the following issues before it can be considered to publish in this journal.

1. The abstract is too long and has many repeated information. After reading it, I still don’t understand what problem the authors plan to solve, and what knowledge gap the authors plan to fill in this research. There are many grammatical errors and contradictions throughout the section. For example, in Line 104, the authors wrote one advantage of FDM is “variety of materials”. Later in Line 110, the authors mentioned that main disadvantage of FDM is the “limited range of appropriate materials”.

2. Line 303, the simulation images for the seven materials should all be provided.

3. Line 327, what is the definition of “strength limit”. The values of strength limits for seven materials should be provided.

4. What is the definition of "resistance to compressive load"? It seems the definition in Line 339 and Line 398 are different. One is closely related to the material, while another one is not affected by the material a lot.

Author Response

Reviewer 1

English language and style

( ) English very difficult to understand/incomprehensible
(x) Extensive editing of English language and style required
( ) Moderate English changes required
( ) English language and style are fine/minor spell check required
( ) I don't feel qualified to judge about the English language and style

Thank you for your opinion.

The English language of the article was corrected by MDPI Language Editing Services.

Yes

Can be improved

Must be improved

Not applicable

Does the introduction provide sufficient background and include all relevant references?

( )

( )

(x)

( )

Are all the cited references relevant to the research?

( )

(x)

( )

( )

Is the research design appropriate?

( )

(x)

( )

( )

Are the methods adequately described?

( )

(x)

( )

( )

Are the results clearly presented?

( )

( )

(x)

( )

Are the conclusions supported by the results?

( )

(x)

( )

( )

Comments and Suggestions for Authors

This manuscript fabricated orthopedic shoes using three additive manufacturing technologies (FDM, SLS, SLA) and seven different materials: thermoplastic copolyester (TPC), polylactic acid (PLA), acrylonitrile butadiene styrene (ABS), polyamide 12 (PA12) powder, polyethylene terephthalate glycol (PETG), photopolymer, polyamide filaments (NYLON). The authors then performed simulation to study the stress and displacement distribution of these seven materials during compressing test. In addition, the authors also performed experimental compression test of these seven materials. The manuscript needs to address the following issues before it can be considered to publish in this journal.

Remark (1): The abstract is too long and has many repeated information. After reading it, I still don’t understand what problem the authors plan to solve, and what knowledge gap the authors plan to fill in this research. There are many grammatical errors and contradictions throughout the section. For example, in Line 104, the authors wrote one advantage of FDM is “variety of materials”. Later in Line 110, the authors mentioned that main disadvantage of FDM is the “limited range of appropriate materials”.

Answer:

The abstract has been corrected.

Thank you for your comment about [17] and [19]. To select materials and suitable printing methods for our research, we analyzed other researchers' experiences and various opinions. We agree with you that by citing sources 17 and 19 we have not fully announced the context of the authors' thoughts. Publication 19 discussed the limitations of commercial FDM printers when considering their application to mass production. To add clarity to the context, we have changed the sentence to read:

“On the other hand, commercial 3D printers suitable for larger-scale production can often only print ABS or PLA, therefore, there are opinions that this 3D printing method is a limited range of appropriate materials [19]”.

Remark (2): Line 303, the simulation images for the seven materials should all be provided.

Answer: In the text, we mentioned that the simulation images for the seven materials variants are visually the same.

In order not to overload the article with excessive information, we only provided an example with one material. s proof that it is repeated in all cases, we the most dangerous areas additionally showed in the images of another material.

Since only the von Misses and displacement results differ, the maximum values of which are analyzed in Figure 7, we think that adding more pictures is not appropriate.

Remark (3): Line 327, what is the definition of “strength limit”. The values of strength limits for seven materials should be provided.

Answer: The text is supplemented:

“However, the consequent stresses are significantly lower than the ultimate strength of all tested materials (table 1), or yield strength values of plastics used in the theoretical simulations works of other researchers: 20 MPa for ABS [49], 45 MPa for PLA [50], 49.7 MPa for PETG [51], 82 MPa for NYLON [52]. Therefore, no plastic deformation will occur in the areas with the greatest impact of heels.”

Remark (4): What is the definition of "resistance to compressive load"? It seems the definition in Line 339 and Line 398 are different. One is closely related to the material, while another one is not affected by the material a lot.

Answer:

For the sake of clarity, we made some corrections to the text:

“The obtained results of theoretic simulation show that the spatial design of the designed heels for orthopedic shoes is the main factor that determines their resistance to compressive loads up to 3000 N, while the influence of the properties of the used materials is significantly smaller.”

While designing the product and performing a theoretical simulation of the compression loads effects, we checked the theoretically possible loads of the heel up to 3000 N.

Among the results, TPC stands out the most, the maximum stress values obtained by other materials at the most dangerous points of the product are relatively close. According to these results, changing the materials in the simulation program did not have a significant impact, all variants should be suitable and not broken during use.

We saw how true it was during the experimental study.

The compressive strength of all variants up to 3000N is very close to that of the theoretical simulation, with only a small displacement difference of up to 0.5mm visible.

However, as the load continues to increase, their behavior clearly differs.

First of all, it is clear that the 3D printing method is a very important factor in the compression resistance of the product due to the different methodic of forming the product.

The results of the experiment also showed that TPC is not suitable for a product of this design and purpose. Although the theoretical displacements in the hazardous areas were the largest of all materials, it was less than 2 mm (less 1 mm at 1000 N), so it was decided to try to make it from this plastic as well.

When comparing four versions of heels produced by the same FDM method and the same printer, obvious differences in compression resistance become apparent when the compression force exceeds 3000N, so the influence of the properties of the plastic itself is already visible here.

Authors:

Thank You very much for your comments and suggestions on the article.

Reviewer 2 Report

The work was interesting, and using different manufacturing techniques is the key highlight of this research work.

1.       The authors have done enough experimental work, but the failure analysis at the micro level is missing in this manuscript.

2.       Some optical and electron microscope study has to be performed in detail to explain the failure analysis.

3.       The Virtual static simulation results need proper validation with analytical calculation.

Author Response

Reviewer 2

English language and style

( ) English very difficult to understand/incomprehensible
( ) Extensive editing of English language and style required
(x) Moderate English changes required
( ) English language and style are fine/minor spell check required
( ) I don't feel qualified to judge about the English language and style

Thank you for your opinion.

The English language of the article was corrected by MDPI Language Editing Services.

Yes

Can be improved

Must be improved

Not applicable

Does the introduction provide sufficient background and include all relevant references?

(x)

( )

( )

( )

Are all the cited references relevant to the research?

(x)

( )

( )

( )

Is the research design appropriate?

(x)

( )

( )

( )

Are the methods adequately described?

( )

(x)

( )

( )

Are the results clearly presented?

( )

(x)

( )

( )

Are the conclusions supported by the results?

( )

(x)

( )

( )

Comments and Suggestions for Authors

The work was interesting, and using different manufacturing techniques is the key highlight of this research work.

Remark (1): The authors have done enough experimental work, but the failure analysis at the micro level is missing in this manuscript.

Answer:

The manuscript is supplemented by several images of damages of FDM-printed heels.

We did not perform failure analysis at the micro level because no damage was detected in the relevant load limits up to 3000 N for this product. It is also visible in the experimental compression curves of the prototypes.

Remark (2): Some optical and electron microscope study has to be performed in detail to explain the failure analysis.

Answer:

According to the obtained experimental curves, in the case of PA12 and photopolymer heels 3D-printed using SLA and SLS methods, it was clear that there was not purposeful in looking for damage at the micro level. 

In all cases of FDM-printed heels, compressed prototypes began to break at loads significantly higher than realistically possible during the production and wearing of orthopedic shoes.

On the other hand, during the experiments, it was observed that all of the cracks of these samples corresponded to the tendencies observed in the work of other researchers: the cracks were caused by the separation of fused filaments.

In further work, long-term cyclic fatigue tests will be performed, so optical microscopy studies will be really relevant.

Remark (3): The Virtual static simulation results need proper validation with analytical calculation.

Answer:

At this stage of research, we did not aim to create a model for analytical calculations.

The purpose of this part of the research was to test different 3D printing methods and select suitable polymer materials suitable for the production of specially designed orthopedic footwear heels. It was important to verify the effect of realistically possible loads on the created spatial structure and the influence of the nature of the materials on the behavior of such a heel during compression.

Authors:

Thank You very much for your comments and suggestions on the article. Your comments will be useful for further research.

Reviewer 3 Report

The paper discusses the Influence of the Type of Plastic and Printing Technologies on the Compressive Behavior of 3D Printed Heel Prototypes. In this study, the authors used seven variants of heels with different polymeric materials using SLS, SLA, and FDM printing techniques. These are PA12 materials by SLS & SLA method; PLA, TPC, ABS, PETG, and NYLON materials by the FDM method. After monitoring the experimental-based compressive behavior, it is further validated by modeling. Where authors suggested traditional wooden heels with hand-made personalized orthopedic footwear can be replaced with good quality PA12. Overall, the manuscript seems good to publish but some minor revisions need to be addressed in the current form of the manuscript. These are:

·       Abstract should revise to add key findings in the context of compression load-bearing value.

·       In the introduction section, the citation is not used as per journal format such as “Rohit Agrawal reviewed the materials….” As on Page 5. These changes need to be addressed thoroughly.

·       In the material and methods section, specify what are the constant parameters while performing experimentation and modeling, respectively.  

·       Future scope could be elaborated in detail.

Hence, reframing of the manuscript should be done.

Author Response

Reviewer 3

English language and style

( ) English very difficult to understand/incomprehensible
( ) Extensive editing of English language and style required
( ) Moderate English changes required
(x) English language and style are fine/minor spell check required
( ) I don't feel qualified to judge about the English language and style

Thank you for your opinion.

The English language of the article was corrected by MDPI Language Editing Services.

Yes

Can be improved

Must be improved

Not applicable

Does the introduction provide sufficient background and include all relevant references?

( )

( )

(x)

( )

Are all the cited references relevant to the research?

( )

(x)

( )

( )

Is the research design appropriate?

(x)

( )

( )

( )

Are the methods adequately described?

( )

(x)

( )

( )

Are the results clearly presented?

(x)

( )

( )

( )

Are the conclusions supported by the results?

(x)

( )

( )

( )

Comments and Suggestions for Authors

The paper discusses the Influence of the Type of Plastic and Printing Technologies on the Compressive Behavior of 3D Printed Heel Prototypes. In this study, the authors used seven variants of heels with different polymeric materials using SLS, SLA, and FDM printing techniques. These are PA12 materials by SLS & SLA method; PLA, TPC, ABS, PETG, and NYLON materials by the FDM method. After monitoring the experimental-based compressive behavior, it is further validated by modeling. Where authors suggested traditional wooden heels with hand-made personalized orthopedic footwear can be replaced with good quality PA12. Overall, the manuscript seems good to publish but some minor revisions need to be addressed in the current form of the manuscript.

These are:

Remark (1): Abstract should revise to add key findings in the context of compression load-bearing value.

Answer: The summary has been corrected.

Remark (2): In the introduction section, the citation is not used as per journal format such as “Rohit Agrawal reviewed the materials….” As on Page 5. These changes need to be addressed thoroughly.

Answer: The text was corrected.

Remark (3): In the material and methods section, specify what are the constant parameters while performing experimentation and modeling,

Answer:

We apologize if we misunderstood your comment. The text is supplemented with a sentence:

“The 50 N preload was set for all tests.”

Parameters that are specified in the text:

In the section “2.3. Static simulation” Figure 3 shows the all data of the finite element mesh used for the simulation; the loads used for the theoretical simulation (1000 N, 2000 N, and 3000 N) are indicated in the text of this part. In the section "2.1. Compression test" the compression speed of 5 mm/min, which was used in the experiment tests, is indicated.

Remark (4): Future scope could be elaborated in detail.

Answer: The text was supplemented.

Authors:

Thank You very much for your comments and suggestions on the article.

Round 2

Reviewer 1 Report

The manuscript has been revised with satisfaction.